# Active Ultrasonic Structural Health Monitoring Enabled by Piezoelectric Direct-Write Transducers and Edge Computing Process

**DOI:** 10.3390/s22155724

**Published:** 2022-07-30

**Authors:** Voon-Kean Wong, Sarbudeen Mohamed Rabeek, Szu Cheng Lai, Marilyne Philibert, David Boon Kiang Lim, Shuting Chen, Muthusamy Kumarasamy Raja, Kui Yao

**Affiliations:** 1Institute of Materials Research and Engineering (IMRE), Agency for Science, Technology and Research (A*STAR), Singapore 138634, Singapore; wong_voon_kean@imre.a-star.edu.sg (V.-K.W.); sc-lai@imre.a-star.edu.sg (S.C.L.); marilyne_philibert@imre.a-star.edu.sg (M.P.); david_lim@imre.a-star.edu.sg (D.B.K.L.); slint.st@gmail.com (S.C.); 2Institute of Microelectronics (IME), Agency for Science, Technology and Research (A*STAR), Singapore 138634, Singapore; sarbudeenmr@ime.a-star.edu.sg (S.M.R.); raja@ime.a-star.edu.sg (M.K.R.)

**Keywords:** structural health monitoring, ultrasonic guided wave, piezoelectric transducer, edge computing

## Abstract

While the active ultrasonic method is an attractive structural health monitoring (SHM) technology, many practical issues such as weight of transducers and cables, energy consumption, reliability and cost of implementation are restraining its application. To overcome these challenges, an active ultrasonic SHM technology enabled by a direct-write transducer (DWT) array and edge computing process is proposed in this work. The operation feasibility of the monitoring function is demonstrated with Lamb wave excited and detected by a linear DWT array fabricated in situ from piezoelectric P(VDF-TrFE) polymer coating on an aluminum alloy plate with a simulated defect. The DWT array features lightweight, small profile, high conformability, and implementation scalability, whilst the edge-computing circuit dedicatedly designed for the active ultrasonic SHM is able to perform signal processing at the sensor nodes before wirelessly transmitting the data to a remote host device. The successful implementation of edge-computing processes is able to greatly decrease the amount of data to be transferred by 331 times and decrease the total energy consumption for the wireless module by 224 times. The results and analyses show that the combination of the piezoelectric DWT and edge-computing process provides a promising technical solution for realizing practical wireless active ultrasonic SHM system.

## 1. Introduction

Structural health monitoring (SHM) involves sensing, collecting, and analyzing data of a structure periodically to determine its conditions over times [1,2]. As a non-intrusive monitoring technique, it is performed on important assets such as buildings [3], bridges [4], pipes [5], railway tracks [6], aircrafts [7], and maritime structures [8] to facilitate condition-based preventive maintenance. The lifetime, reliability, and safety of these structures are therefore greatly improved under long-term fatigue and corrosive working environments. Sensors play an important role in an SHM system. These sensors, also known as sensing nodes, are usually installed on major hotspots or safety-critical components. The multitude of these sensors collectively form a spatially distributed sensing network, capable of conducting extensive monitoring of the structure’s integrity [9,10,11]. With the Internet of Things (IoTs) technologies, the sensed data acquired by the nodes can be wirelessly transmitted to a remote base station or secured cloud sever for collation and analysis [12,13,14]. The modern SHM technology is emerging an important element of Industry 4.0, as it greatly enhances the efficiency of management for infrastructure and valuable assets with the desired automation of routine structural inspection activities. 

Ultrasonic waves can be utilized as the effective structural integrity indicators for realizing SHM in passive and active ways. The passive method solely measures the mechanical parameters, such as the vibration or acoustic emission of the structure to assist in the diagnostic process without exerting additional excitation energy on the structure [15]. The active method typically involves generating ultrasonic excitation on the structure, and detecting the ultrasonic waves scattered by the boundaries, obstructions, or defects [16,17]. With the advantages of the selected working frequency and high signal-to-noise ratio, the active ultrasonic method is capable of detecting minute defects down to submillimeter sizes and is highly versatile for different applications including identification of fine cracks, monitoring of corrosion level, measurement of structural thicknesses, and assessment of adhesive joint qualities [18,19,20].

The conventional active ultrasonic SHM methods are based on the bulk waves propagating in the thickness direction. This method is usually confined to the detection of localized defects [21]. This limitation, however, has been overcome with the use of ultrasonic guided wave (UGW) such as Lamb wave, which is a kind of dispersive wave travelling along the structural boundaries [22,23]. As the wave energy dissipates gradually in the course of propagation, UGW can travel over certain distances along the structure. In conjunction with its ability to access obstructed areas, UGW enables active SHM to be realistically performed on enormous and complex structures [24,25]. 

Active ultrasonic SHM system typically makes use of multiple lead zirconate titanate (PZT) piezoelectric ultrasonic transducers for ultrasonic excitation and detection in the targeted structure [26]. However, the use of such ceramic-based piezoelectric ultrasonic transducers is accompanied by a few major problems. First, the discrete ceramic-based piezoelectric ultrasonic transducers are usually bulky and possess poor surface conformability, in turn limiting their application on structures with space and weight constraints, as well as those of complex surface topologies. Hence, it is often not practical to implement these ceramic-based piezoelectric ultrasonic transducers in large arrays over a wide-area structure. Recent cost–benefit study shows that it is economically impractical to implement SHM using the bulky piezoelectric ultrasonic sensors to replace manual inspection if the weight and cost cannot be reduced [27]. Secondly, the transducers must be bonded onto the structure using adhesives agents, such as epoxies. These adhesives with the massive transducer loads are susceptible to degradation under cyclic temperature and vibration due to interfacial stresses [28,29]. This can eventually lead to delamination or even cracking of the transducer, thus limiting the durability and reliability of the active ultrasonic SHM system. Thirdly, the manual installation process of the bulky discrete transducers results in variations in terms of position and alignment of the transducers, and the thickness and uniformity of the adhesives, and thus the inconsistencies in the acoustic coupling effect between the transducers and structures [30]. Fourthly, the PZT material contains a toxic lead compound, which causes health and environmental concerns.

One strategy for implementing sustainable, reliable, and cost-effective active ultrasonic SHM technology is to produce transducers directly on the structure through a batch processing method, such as fabricating piezoelectric ultrasonic transducers through a coating and patterning process for realizing active ultrasonic SHM [30,31,32,33]. The resulting ultrasonic transducers, also known as the direct-write transducers (DWT), comprises a functional piezoelectric film highly conformal to the underlying surface topologies, with a minimized impact on the overall structural weight and dimension. The application of DWTs for active SHM has been demonstrated on metallic plates, pipes, and carbon fiber reinforced polymer plates [34,35,36,37,38]. By virtue of the above advantages, DWT technology enables vast quantities of transduction elements in the form of arrays to be feasibly fabricated on a wide range of structures. 

To scale up active ultrasonic SHM into a wide-area sensing network (WSN), further challenges associated with electrical wiring and circuitry, power installation, and signal transmission of the sensing nodes have to be addressed [39,40]. Various engineering solutions, including wireless data transmission, low-power electronics sustainable by batteries or energy harvesters and circuitry miniaturization, have been applied for SHM [41,42,43]. The desired outcome is a highly compact, portable, and battery-powered or even self-powered sensing node that can be deployed in large quantities simultaneously, with each sensing node connected wirelessly to a central data management station. Some advancements of WSN, notably for active diagnostic of large structures, have been reported. For instance, a lightweight and compact multi-response based WSN network system for impact monitoring using 84 PZT ultrasonic sensors aided with a smart analytical algorithm is reported, achieving an impressive impact localization accuracy of 96% [44]. 

Another challenge faced by WSN pertains to the massive raw data acquired by numerous sensing nodes to be transmitted wirelessly in real time to the base station, which can potentially lead to bandwidth congestion. This problem is accentuated by the pressing demand for denser network involving more sensing nodes, higher detection resolution and faster sampling rate over longer sensing durations for improved data quality. Thus, edge-computing is desired for processing the raw sensing data into intermediary or end results of much smaller data sizes within the sensing node prior to transmission to the base station [45,46]. With the edge-computing, the sensing nodes can be developed on advanced computational resources capable of executing the analytical algorithm at high speed for real time SHM. 

To keep up with the proliferation of sensing elements and sensed data, the paper hereby proposes a wireless solution and edge computing to be combined with the DWT, with the aim at realizing a sensing network towards practically feasible active ultrasonic SHM technology in the future. For operation demonstration of an active ultrasonic SHM edge computing system with a DWT array, a linear DWT array is designed and fabricated on an aluminum alloy plate with simulated defect in this work. An active ultrasonic SHM edge computing circuit and system with wireless data transfer capability is designed and implemented. A sequential ultrasonic scan is performed using the active ultrasonic SHM edge computing system with the DWT array, and an ultrasonic image is formed in our experimental demonstration. 

## 2. Active Ultrasonic SHM Edge Computing System with DWT Array

The modern SHM technology is advancing towards the use of highly distributed wireless sensors with edge intelligence to process massive sensing data. Accordingly, as illustrated in Figure 1, an active ultrasonic SHM system operating on UGW based on DWT arrays fabricated in situ on the host structure is proposed. DWT array can be batch formed with multiple DWT elements by automated processing, which can enhance the implementation scalability. Furthermore, the lightweight, small profile and high conformability features of the DWT array allow it to be fabricated in situ on the structure with limited space and complex geometry. Each DWT is configurable as an ultrasonic actuator or sensor by interchangeably connecting to an electrical ultrasonic pulser or analog signal receiver subsystem, respectively. The DWTs thus provide the dual functionalities of ultrasonic pulse excitation and ultrasonic signal detection.

Data sensing and processing are both performed within the edge computing system. The raw data of the detected ultrasonic signals are processed into results indicative of the current structural health condition. Thereafter, the processed ultrasonic signals are transmitted wirelessly, with a potentially greatly reduced amount of data at reduced energy consumption, to a secured cloud platform accessible by an SHM data monitoring center. An active ultrasonic SHM edge computing system capable of real-time sensing and in situ data analysis has thus been realized with the potential to be scaled-up into an Internet-of-things (IoTs) network for monitoring various kinds of host structures.

## 3. Materials and Methods

### 3.1. Design and Fabrication of Linear DWT Array

For the operation demonstration, an aluminum alloy plate with the dimensions of 300 mm × 300 mm × 1.6 mm with a simulated defect with the dimensions of 45 mm × 2 mm × 1 mm is used, as shown in Figure 2a. A linear ultrasonic transducer array comprising eight DWTs is implemented. 

The DWTs are designed with a comb-shaped top electrode for enhanced Lamb wave directivity and mode selectivity. The Lamb wave S2 mode excited at 3.2 MHz is selected for this work. The phase velocity and group velocity dispersion curves of a 1.6 mm-thick aluminum alloy plate given in Figure 3 are referred to aid the design for the comb-shaped top electrode. The wavelength of the S2 mode is calculated to be 2.4 mm by dividing the phase velocity by the corresponding frequency. To effectively generate and detect S2 mode at 3.2 MHz, the width of the comb-shaped top electrode is selected to be 1.2 mm accordingly, in order for the electrode periodicity to match one S2 mode wavelength. The generated S2 mode has a group velocity of 3700 m/s in the aluminum alloy plate.

The in-house fabricated DWTs consist of a piezoelectric coating made of poly(vinylidene fluoride-trifluoroethylene) (P(VDF-TrFE)) copolymer, silver electrode, and a protective layer, as illustrated in Figure 2b. By using an aerosol spray coating system (Nordson in Westlake, OH, USA), the piezoelectric polymer coating with a thickness of about 13 μm was spray-coated on the aluminum alloy plate at 80 °C. The P(VDF-TrFE) coating was then annealed at 135 °C for 30 min by applying heat to the structure. Next, the P(VDF-TrFE) coating was poled by a noncontact corona discharge gun. A layer of insulating sticker was adhered next to the piezoelectric coating. This layer is for the direct-write signal traces, which connect the DWTs to a customized flexible printed circuit (FPC) with eight signal channels. The customized FPC was terminated to a D-Subminiature (DB9) connector, which was then be connected to the circuit for active ultrasonic edge computing. Thereafter, a sticker mask was used for deposition and patterning of the top silver electrode and signal traces, followed by curing at 70 °C. Finally, a protective coating made of epoxy adhesive (Araldite 2011) was applied on the DWT. 

An ultrasonic image can be obtained using the linear DWT array as fabricated above by performing sequential ultrasonic scan. This is done by activating one DWT element at a time, sequentially, operating in pulse-echo mode. In pulse-echo mode, the selected DWT element will generate S2 mode and detect ultrasonic signals reflected from the edge of the aluminum alloy plate and from the edge of the simulated defect. 

### 3.2. Design and Implementation of the Active Ultrasonic SHM Edge Computing Circuit and System

Our in-house designed active ultrasonic edge circuit is made of three modules: ultrasonic transmit (USTx) module, ultrasonic receive (USRx) module, and wireless module, as shown in Figure 4. The main component in the USTx module is an ultrasonic pulser chip (MD2131) for the generation of ultrasonic signals. The USTx module is connected to the linear ultrasonic transducer array. Channel selection for the linear transducer array is performed via an 8-channel high voltage analog switch (HV2918). An active transmit-receive (Tx/Rx) switch (MD0101, Microchip Technology in Chandler, AZ, USA) is used for preventing high voltage transmit pulses from entering the USRx module. An analogue front-end (AFE) integrated chip (AFE5812, Texas Instruments in Dallas, TX, USA) is used in the USRx module for digitization of the ultrasonic signals. The AFE5812 contains a programmable low-noise amplifier (LNA), and a 14-bit 40 MSPS analogue-to-digital converter (ADC). The edge-computing capability is enabled using an on-board microcontroller (MCU) (STM32F103, STMicroelectronics in Singapore) and a field-programmable gate array (FPGA) module (TE0714-03-35-2I XC7A50T, Trenz Electronics in Hüllhorst, Germany), which is equipped with a 50 K gates FPGA for controlling both ultrasonic pulser chip and the analogue front-end chip. In the edge computing circuit, the MCU acts as a master controller that can switch the entire system off to conserve energy when the scan is not being performed. The MCU is also used to control the wireless module (ZigBee, Chengdu Ebyte Electronic Technology in Sichuan, China). The entire ultrasonic edge system is powered by a 3.7–4.2 VDC battery, which powers up each module on-board. Appropriate voltage regulators are used to step-down and step-up the DC voltages for various modules. Load switch chips are used to implement power up and power down of modules in the edge circuit to optimize the power consumption. 

A flow chart describing the functionality of the developed active ultrasonic edge system, including edge-computing processes, is illustrated in Figure 5. The active ultrasonic edge system is configurable with a host device through the ZigBee wireless link. This host device can be remotely accessed from a SHM data monitoring center. Parameters for the ultrasonic pulser such as burst amplitude and frequency can be reconfigured when needed by programming the MCU. The ultrasonic pulser is set to generate a Hanning-windowed sine-wave excitation signal with peak-to-peak amplitude of ~80 Vpp and frequency of 3.2 MHz. Next, the MCU configures the AFE chip parameter with 40 MSPS and 14 bits/sample. Once the parameters are set, the MCU activates the corresponding DWT-element through switching the 8-channel high voltage analog switch. An excitation signal is then generated by the ultrasonic pulser and an ultrasonic wave is generated by the selected DWT-element. After the excitation is completed, AFE starts receiving the ultrasonic signals and stores in an array size of 4 K deep first-in first-out (FIFO) memory inside the FPGA. Once the FIFO memory is full, the MCU reads all the 4 K data and stores it in its internal memory. 

The raw ultrasonic signal is processed via edge-computing processes using the MCU. The raw ultrasonic signal is first processed by a finite impulse response (FIR) Band pass filter to remove power supply and system noises. The filtered ultrasonic signal is then rectified by obtaining absolute values of the filtered ultrasonic signal. Next, the rectified ultrasonic signal is enveloped and then decimated by a factor of 40 with an FIR filter to down sample the enveloped ultrasonic signal from 4000 number of data samples to 100 number of data samples. The decimated ultrasonic signal is then transmitted wirelessly to the host device through a ZigBee wireless link. The system repeats this process until the sequential ultrasonic scan process is completed. Thereafter, the edge-processed ultrasonic signals are uploaded to a cloud server for enabling active ultrasonic SHM applications. The active ultrasonic edge system is powered down and waits for the next sequential scan. 

## 4. Results 

### 4.1. Active Ultrasonic Testing Results

A sequential ultrasonic scan was performed by conducting pulse-echo testing using one DWT element as the ultrasonic pulser at a time. The ultrasonic pulser was set to generate a Hanning-windowed, three cycle sine-wave excitation signal with peak-to-peak amplitude of ~80 Vpp and frequency of 3.2 MHz. The excitation signal in both time and frequency domains are presented in Figure 6. S2 mode Lamb wave at 3.2 MHz was generated by the selected DWT element.

The ultrasonic signals detected by the DWTs were digitized by the AFE and stored in the FPGA, then later read by MCU to its internal memory. The ultrasonic signals were processed by the MCU following the edge-computing processes as described in Figure 5. As an example, the raw ultrasonic signal generated and detected by the 8th DWT element is presented in Figure 7a. In the band pass filtered ultrasonic signal, Figure 7b, the S2 mode reflected from edge of the plate can be observed at *t* = 71 µs. The large signals at *t* < 10 µs were produced by the electromagnetic interference (EMI). The filtered ultrasonic signals were then rectified and enveloped, as in Figure 7c, which is thereafter down sampled by factor of 40 using a decimating FIR filter, as in Figure 7d. 

The processed ultrasonic signals were then transferred to the host device wirelessly via the ZigBee wireless link. Once all eight ultrasonic signals were received by the host device, an ultrasonic image can be generated, as presented in Figure 8. The ultrasonic image contains the decimated ultrasonic signal from the DWTs. The distance from the center of the DWT to the edge of the simulated defect and edge of the aluminum alloy plate are 60 mm and 130 mm, respectively, as illustrated in Figure 8a. The calculated arrival time for the S2 mode reflected from edge of the simulated defect and edge of the aluminum plate are 32 µs and 70 µs, respectively. From the ultrasonic image, given in Figure 8b, the S2 mode reflected from the edge of the simulated defect and the edge of the aluminum alloy plate can be clearly observed. The S2 mode reflected from the edge of the simulated defect can only be observed from DWT-element 3 to 6, which demonstrated the capability of defect imaging using the active ultrasonic SHM edge system with the DWT array. 

### 4.2. Performance Enhancement Enabled by Edge-Computing Processes

The performance of the wireless module for the active ultrasonic SHM edge device is evaluated based on the total time needed for data transfer and total energy consumption per sequential scan. One DWT element is configured to one pulse-echo mode at a time during sequential scan, which produces the total number of eight ultrasonic scans in a complete sequential scan routine. A comparison of energy saving capability per sequential scan enabled by edge-computing in the wireless module is tabulated in Table 1. Without enabling edge-computing, a large amount of data is generated for a complete sequential scan routine (3640 kbits). This is due to the high sampling rate demanded to digitize the ultrasonic signal containing S2 mode. After the edge-computing processes, the data generated per sequential scan routine have been reduced substantially to 11 kbits. 

The Zigbee wireless link at 2.4 GHz has a data transfer rate of 250 kbps. By enabling edge-computing, the total data transfer time was decreased significantly from 28,437 ms to 89 ms. Furthermore, the total energy consumption of the ZigBee RF transceiver module in the active ultrasonic SHM edge system has decreased from 3540 µJ to 15.8 µJ by introducing the edge-computing processes. Once the data size decreased, data transfer time as well as the total energy consumption is decreased correspondingly.

## 5. Discussion

Active ultrasonic SHM is one of the most promising technologies to replace current manual inspection in many applications. This is attributed to its superior sensitivity to structural damages and increased monitoring range enabled by UGW [22,25]. However, several challenges unsolved have prevented the practical applications of active ultrasonic SHM for many years, particularly including large weight and cost of the ultrasonic transducers, cables, and instrumentations [22,27], as described in the Introduction section. 

In this work, lightweight and low-profile DWTs that can be batch-produced through scalable coating process are proposed instead of manual installation of discrete bulky ultrasonic transducers. This work has also addressed heavy cabling issue by introducing direct-write conductive signal trace that connects to a low-profile customized FPC. Both the direct-write transducers and signal traces are highly conformal and can be fabricated in situ on the surfaces with complex geometry. The customized FPC is flexible in nature and can be folded and bonded on surface of a structure. By using wireless data transfer, weight from long cabling is significantly reduced. 

Another barrier for realizing WSN for active ultrasonic SHM is related to the massive amount of raw data acquired by the sensing nodes due to the high sampling frequency requirement for receiving the ultrasonic signals [22,47]. For example, in this work, sampling rate of 40 MSPS is implemented in order to capture S2 mode at 3.2 MHz. Depending on the ultrasonic wave propagation distance and its velocity, the data size of ultrasonic signal can be huge. Large area monitoring will translate to longer data transfer time and higher energy consumption. Hence, edge computing is essential so that the raw data could be processed at the sensor node before being transferred to the base station. In this work, the DWT array is connected to the edge computing circuit dedicatedly designed for realizing active ultrasonic SHM, in which ultrasonic signals can be processed by the on-board edge-computing processes before wirelessly transmitting to a remote host device. The amount of data transferred has greatly reduced by 331 times by introducing edge-computing processes to the sensor node. 

Energy consumption is another important factor to be considered for SHM system [39,40]. By decreasing data size for the ultrasonic signals through the edge-computing processes, the total energy required for a sequential ultrasonic scan has been substantially decreased by 224 times. Parameters such as number of signal average, ultrasonic signal duration, and FIR decimation can be further tuned to reduce total energy consumption per ultrasonic scan. 

The current prototype version is made of commercial off-the-shelf integrated chips. For future development, the size, weight, and energy consumption of the active ultrasonic SHM edge system can be further reduced by using system-on-chip (SoC) design. Once energy consumption is further reduced, autonomy of the active ultrasonic SHM edge device can be realized by integrating energy harvesters into the system [48].

## 6. Conclusions

An active ultrasonic SHM technology enabled by DWT array and edge computing process is proposed. The operation feasibility of the monitoring function is demonstrated with Lamb wave excited and detected by a linear DWT array fabricated in situ from piezoelectric P(VDF-TrFE) polymer coating on an aluminum alloy plate with a simulated defect. The DWT array features lightweight, small profile, high conformability, and implementation scalability, whilst the edge-computing circuit dedicatedly designed for the active ultrasonic SHM is able to perform signal processing at the sensor nodes before wirelessly transmitting the data to a remote host device. The DWT array and edge computing circuit is connected using conformal direct-write signal traces and a customized FPC. The successful implementation of edge-computing processes is able to decrease the amount of data to be transferred from the DWT by 331 times and the total energy consumption by 224 times for the wireless module. Our results and analyses have exhibited that the combination of piezoelectric DWT and edge-computing process is promising for realizing wireless active ultrasonic SHM with improved technical and economic viability, as highly demanded in digital twin and IoT technologies. 

## Figures and Tables

**Figure 1 sensors-22-05724-f001:**
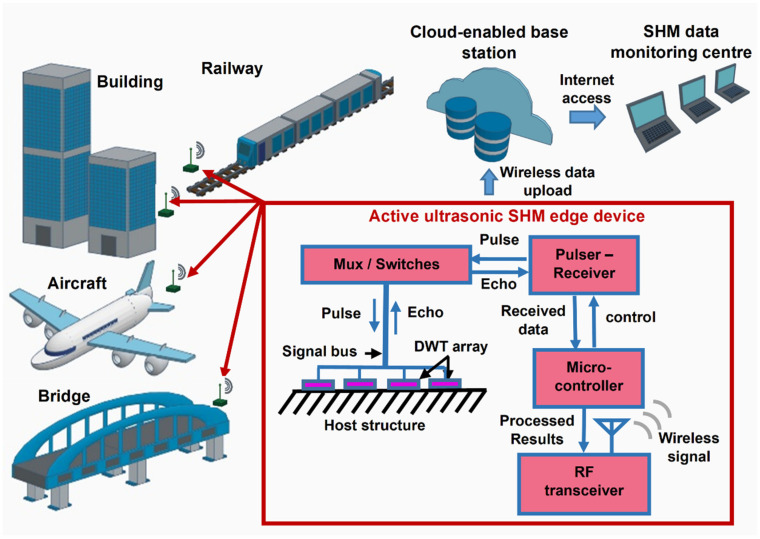
Conceptual illustration of the proposed active ultrasonic SHM edge computing system with DWT arrays in-situ fabricated on various kinds of host structures, scalable for Internet-of-things (IoTs) applications.

**Figure 2 sensors-22-05724-f002:**
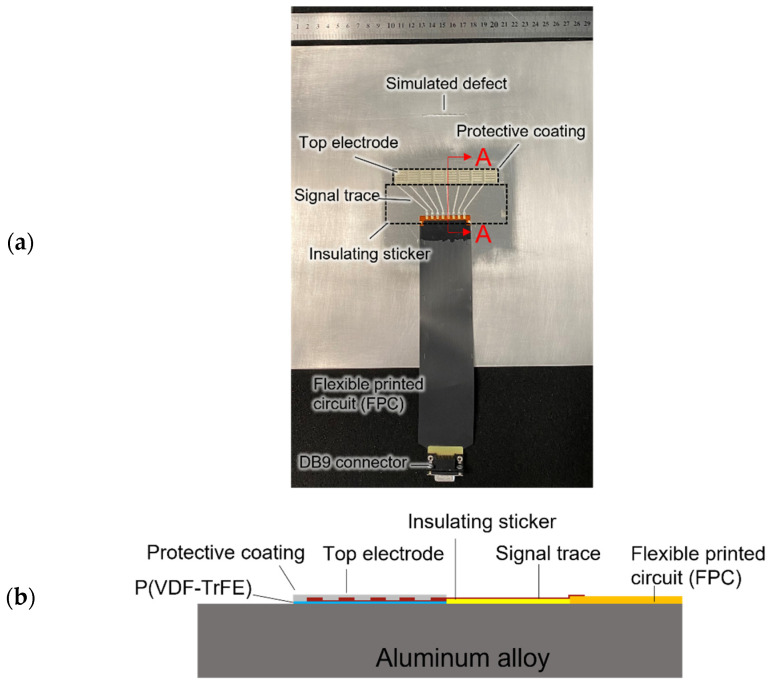
Aluminum alloy plate with a linear direct-write transducer (DWT) array comprising eight transducers: (**a**) photo of the test coupon; and (**b**) sectional view A-A showing layers of the DWT design.

**Figure 3 sensors-22-05724-f003:**
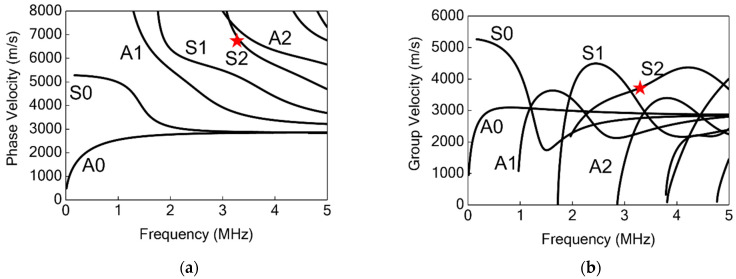
(**a**) Phase velocity and (**b**) group velocity dispersion curves for 1.6 mm-thick aluminum alloy plate. Here, A0, A1, and A2 are the antisymmetric Lamb wave modes; S0, S1, and S2 are the symmetric Lamb wave modes. The red star indicates the selected S2 mode at 3.2 MHz.

**Figure 4 sensors-22-05724-f004:**
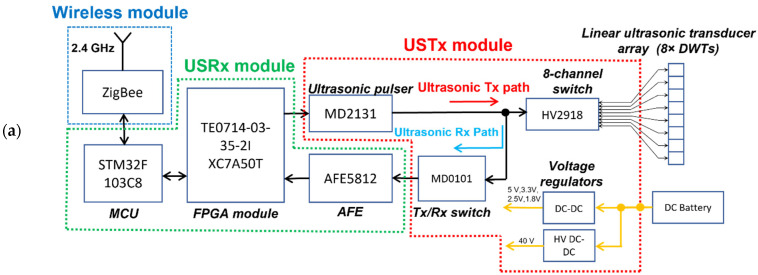
(**a**) Block diagram and (**b**) photo of the modules in the active ultrasonic SHM edge circuit.

**Figure 5 sensors-22-05724-f005:**
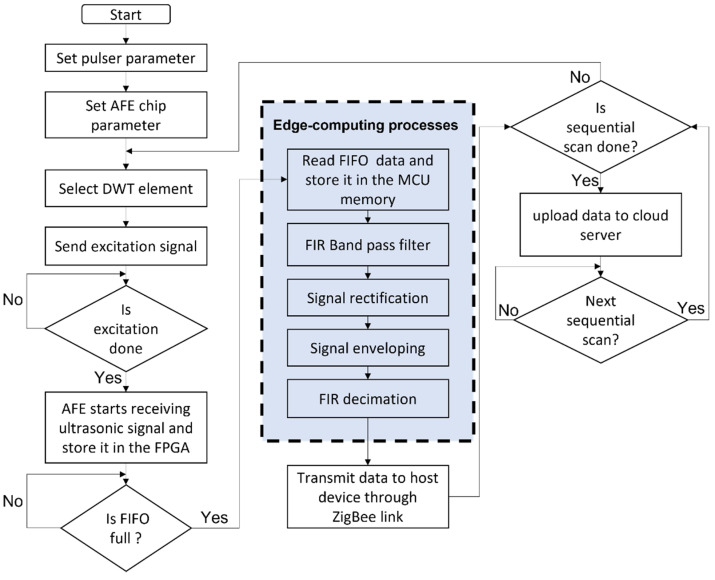
Flow chart of the ultrasonic signal collection, processing, and wireless transmission process, describing the functionality of the developed active ultrasonic SHM edge system, including edge-computing processes.

**Figure 6 sensors-22-05724-f006:**
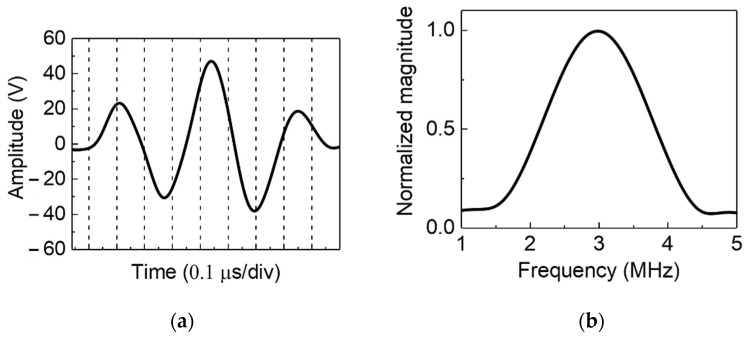
Excitation signal generated by the active ultrasonic SHM edge system: (**a**) time domain, with the vertical dashed lines indicating individual time divisions and (**b**) frequency domain.

**Figure 7 sensors-22-05724-f007:**
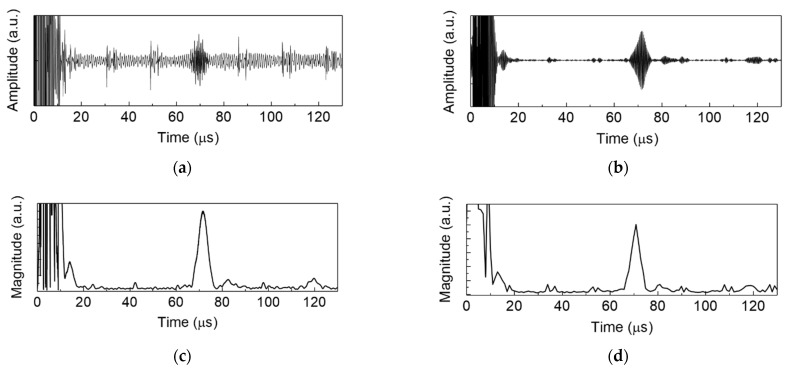
As an example, experimental results for the 8th DWT element operation in the active ultrasonic SHM edge system: (**a**) raw ultrasonic signal; (**b**) band passed filtered signals; (**c**) rectified and enveloped ultrasonic signal; and (**d**) decimated ultrasonic signal.

**Figure 8 sensors-22-05724-f008:**
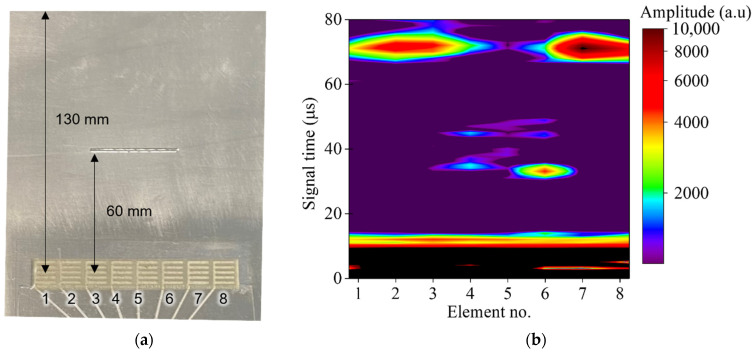
(**a**) Illustration of the ultrasonic wave propagation distance from DWT elements (1–8) to the edge of the aluminum alloy plate and edge of the simulated defect; (**b**) ultrasonic image from sequential ultrasonic scan by conducting pulse-echo ultrasonic testing using one DWT element at a time with the active ultrasonic SHM edge device.

**Table 1 sensors-22-05724-t001:** Comparison of energy savings in the wireless module with and without edge-computing for a complete sequential scan routine.

	Data Generated (kbits)	Data Transfer Time (ms)	Total Energy Consumption (µJ)
Without edge-computing	3640	28,437	3540.0
With edge-computing	11	89	15.8

## Data Availability

Not applicable.

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
