# Peer review of "Active Ultrasonic Structural Health Monitoring Enabled by Piezoelectric Direct-Write Transducers and Edge Computing Process"

_sensors, 2022, doi:10.3390/s22155724_

Round 1

Reviewer 1 Report

The study hereby proposes a wireless solution and edge computing to be combined with the DWT for realizing a sensing network towards practically feasible active ultrasonic SHM technology in the future. The paper is interesting and well-written. The abstract is a concise summary of the manuscript's contents. The introduction introduces the research status and the necessity of research. The results are relevant and well-analyzed in the context of up-to-date literature. I found the study innovative. I recommend to accept this manuscript.

Author Response

Manuscript ID: sensors-1815705

Response to Reviewer 1 Comments

Point 1: The study hereby proposes a wireless solution and edge computing to be combined with the DWT for realizing a sensing network towards practically feasible active ultrasonic SHM technology in the future. The paper is interesting and well-written. The abstract is a concise summary of the manuscript's contents. The introduction introduces the research status and the necessity of research. The results are relevant and well-analyzed in the context of up-to-date literature. I found the study innovative. I recommend to accept this manuscript.

Response 1: Thank Reviewer 1 for the positive comments and recommendation.

Reviewer 2 Report

The paper studies on an active ultrasonic SHM technology enabled by direct-write transducer (DWT) array and edge computing process. In this work, the authors propose a wireless solution and edge computing to be combined with the DWT for realizing a sensing network towards practically feasible active ultrasonic SHM technology. The author concluded that The DWT array features lightweight, small profile, high conformability, and implementation scalability. Even though the content is novel to some extent, there are required revisions as below.

1. It is not clear how the sample dimension and defect size affect the recording of SHM? In this work, they only tested for one specific sample/defect size.

2. Despite providing a schematic sketch of the test prototype, it seems to me that there lack of details step-by-step fabrication/manufacturing. That would make it easy for readers to comprehend.

3. It is still not clear to me that the novelty of this work in terms of new materials, new methods, or new analysis. Could the authors highlight and compare with similar works? What make them different from [16-21]?

Reviewer 3 Report

Abstract

The abstract was well written. However, the following changes should be effected

Lines 24-25: Authors need to clearly state in quantitative form how much data was decreased with the implementation of the edge-computing process

Lines 26-28: Quantitative results could be more presentable

1. Introduction

 Lines 121-124: ''To keep up with the proliferation of sensing elements and sensed data, the paper 121 hereby proposes a wireless solution............''. Authors should clearly state the aims and objectives of the investigation

2. Active Ultrasonic SHM Edge Computing System with DWT Array

If this is a background study, I suggest that this section begins written as ''Background of Active Ultrasonic SHM Edge Computing System with DWT Array''. Also, the section must be expanded if the paper is to be accepted. Any uses? Authors should clarify the relevance of this section.

3. Materials and Methods

Lines 146-149: Authors should justify the use of such dimensions

Author Response

Manuscript ID: sensors-1815705

Response to Reviewer 3 Comments

Point 1: The abstract was well written. However, the following changes should be effected

Lines 24-25: Authors need to clearly state in quantitative form how much data was decreased with the implementation of the edge-computing process

Lines 26-28: Quantitative results could be more presentable

Response 1: We thank the reviewer for this suggestion. Quantitative results are provided in Lines 24-29 of the revised abstract as:

The successful implementation of edge-computing processes is able to greatly decrease the amount of data to be transferred by 331 times and decrease the total energy consumption for the wireless module by 224 times. and thus the total energy consumption by hundreds of times. The results and analyses show that the combination of the piezoelectric DWT and edge-computing process provides a promising technical solution for realizing practical wireless active ultrasonic SHM system.    

Point 2: Introduction

Lines 121-124: ''To keep up with the proliferation of sensing elements and sensed data, the paper  hereby proposes a wireless solution............''. Authors should clearly state the aims and objectives of the investigation

Response 2: Aims and objectives of this work are added in Lines 124-133 of the revised manuscript:

To keep up with the proliferation of sensing elements and sensed data, the paper hereby proposes a wireless solution and edge computing to be combined with the DWT, with the aim atfor realizing a sensing network towards practically feasible active ultrasonic SHM technology in the future. For operation demonstration of an active ultrasonic SHM edge computing system with a DWT array, a linear DWT array is designed and fabricated on an aluminum alloy plate with simulated defect in this work. An active ultrasonic SHM edge computing circuit and system with wireless data transfer capability is designed and implemented. Sequential ultrasonic scan is performed using the active ultrasonic SHM edge computing system with the DWT array, and an ultrasonic image is formed in our experimental demonstration.       

Point 3: Active Ultrasonic SHM Edge Computing System with DWT Array

If this is a background study, I suggest that this section begins written as ''Background of Active Ultrasonic SHM Edge Computing System with DWT Array''. Also, the section must be expanded if the paper is to be accepted. Any uses? Authors should clarify the relevance of this section.

Response 3: This section is relevant to the manuscript as it provides an overview of ourconcept of the active ultrasonic SHM edge computing system with DWT array. Without this section, the reader may not have the big picture on how this proposed technology can be effectively applied for many practical SHM applications. For this purpose, this section is expanded in the revised manuscript in Lines 135-166 as:

The modern SHM technology is advancing towards the use of highly distributed wireless sensors with edge intelligence to process massive sensing data. Accordingly, as illustrated in Fig. 1, we propose an active ultrasonic SHM system operating on UGW based on DWT arrays in-situ fabricated in situ on the host structure is proposed. DWT array can be batch formed with multiple DWT elements by automated processing, which can enhance the implementation scalability. Furthermore, the lightweight, small profile and high conformability features of the DWT array allow it to be fabricated in situ on the structure with limited space and complex geometry. Each DWT is configurable as an ultrasonic actuator or sensor by interchangeably connecting to an electrical ultrasonic pulser or analog signal receiver subsystem, respectively. The DWTs, thus, provide the dual functionalities of ultrasonic pulse excitation and ultrasonic signal detection.

Data sensing and processing are both performed within the edge computing system. The raw data of the detected ultrasonic signals are processed into results indicative of the current structural health condition. Thereafter, the processed ultrasonic signals are transmitted wirelessly, with potentially greatly reduced amount of data at reduced energy consumption, to a secured cloud platform accessible by SHM data monitoring center. An active ultrasonic SHM edge computing system capable of real time sensing and in situ data analysisin-situ has thus been realized with the potential to be scaled-up into an Internet-of-things (IoTs) network for monitoring various kinds of host structures.

Point 4: Materials and Methods

Lines 146-149: Authors should justify the use of such dimensions

Response 4: Thank you for the comment. The large plate surface area is chosen for demonstration of large area monitoring. Meanwhile, the length of the simulated defect is chosen to span from DWT-element 3 to 6 so that the ultrasonic image in Fig. 8 can identify structure’s conditions (with and without defect).

Reviewer 4 Report

The submitted paper (Active Ultrasonic Structural Health Monitoring Enabled by Piezoelectric Direct-write Transducers and Edge Computing Process) is important for the construction industry as it presents a method for structural health monitoring. Generally, this paper is good and can be accepted after the following corrections/changes:

1- There are a few grammar and spelling mistakes. consult an editor to remove them 

2- Avoid mass citation; no need to use more than 3 references per piece of information. For example, you used 4 and 6 references to support some sentences in the introduction, such as [4-8], [9-12] and [22-27].

3- Remove pronounces (such as we) because academic writing uses the third person 

4- It would be a good idea to mention the recent studie on the monitoring techniques, such as:

https://kijoms.uokerbala.edu.iq/cgi/viewcontent.cgi?article=2408&context=home

5- Remove old references (older than 10 years) where possible

Author Response

Manuscript ID: sensors-1815705

Response to Reviewer 4 Comments

The submitted paper (Active Ultrasonic Structural Health Monitoring Enabled by Piezoelectric Direct-write Transducers and Edge Computing Process) is important for the construction industry as it presents a method for structural health monitoring. Generally, this paper is good and can be accepted after the following corrections/changes:

Point 1: There are a few grammar and spelling mistakes. consult an editor to remove them 

Response 1: We thank the reviewer for the comment. Grammar and spelling in the manuscript is checked through with improvement.

Point 2: Avoid mass citation; no need to use more than 3 references per piece of information. For example, you used 4 and 6 references to support some sentences in the introduction, such as [4-8], [9-12] and [22-27].

Response 2: Revisions are made as below in reference to the comment.

[4-8] refer to individual assets, they are referred individually in Lines 35-38 of the revised manuscript as:

As a non-intrusive monitoring technique, it is performed on important assets, such as buildings [3], bridges [4], pipes [5], railway tracks [6], and aircrafts [7], and maritime structures [8] to facilitate condition-based preventive maintenance [4-8].

[9-12] is revised in Lines 42-46 of the revised manuscript as:

The multitude of these sensors collectively form a spatially distributed sensing network, capable of conducting extensive monitoring of the structure’s integrity [9-129-11]. With the Internet-of Things (IoTs) technologies, the sensed data acquired by the nodes can be wirelessly transmitted to a remote base station or secured cloud sever for collation and analysis [13-1512-14].

[22-27] is revised in Lines 56-61 of the revised manuscript as:

With the advantages of the selected working frequency and high signal-to-noise ratio, the active ultrasonic method is capable of detecting minute defects down to submillimeter sizes, and is highly versatile for different applications including identification of fine cracks, monitoring of corrosion level, measurement of structural thicknesses and assessment of adhesive joint qualities [18-2022-27].

Point 3: Remove pronounces (such as we) because academic writing uses the third person

Response 3: We thank the reviewer for pointing this out. The pronounces are revised accordingly.

Point 4: It would be a good idea to mention the recent studie on the monitoring techniques, such as:

 https://kijoms.uokerbala.edu.iq/cgi/viewcontent.cgi?article=2408&context=home

Response 4: Thank you for the suggestion. The mentioned paper is cited as reference [8] in Lines 35-38 of the revised manuscript:

As a non-intrusive monitoring technique, it is performed on important assets, such as buildings [3], bridges [4], pipes [5], railway tracks [6], and aircrafts [7], and maritime structures [8] to facilitate condition-based preventive maintenance [4-8].

Point 5: Remove old references (older than 10 years) where possible

Response 5: In reference to this comments, some older references which are not critically relevant are removed from the manuscript.

Round 2

Reviewer 2 Report

I am satisfied with the answers.